# The Oligotrophy to the UlTra-oligotrophy PACific Experiment (OUTPACE cruise, Feb. 18 to Apr. 3, 2015)

Thierry Moutin[1], Andrea Doglioli[1], Alain De Verneil[1], Sophie Bonnet[2]

[1] Aix Marseille Université, CNRS, Université de Toulon, IRD, OSU Pythéas, Mediterranean Institute of Oceanography (MIO), UM 110, 13288, Marseille, France.
[2] Aix Marseille Université, CNRS, Université de Toulon, IRD, OSU Pythéas, Mediterranean Institute of Oceanography (MIO), UM 110, 98848, Nouméa, New Caledonia.

*Correspondence to*: Thierry Moutin (Thierry.moutin@mio.osupytheas.fr)

**Abstract.** The overall goal of OUTPACE (Oligotrophy to UlTra-oligotrophy PACific Experiment) was to obtain a successful representation of the interactions between planktonic organisms and the cycle of biogenic elements in the western tropical South Pacific Ocean across trophic and $N_2$ fixation gradients. Within the context of climate change, it is necessary to better quantify the ability of the oligotrophic ocean to sequester carbon through biological processes. OUTPACE was organized around three main objectives which were: (1) To perform a zonal characterization of the biogeochemistry and biological diversity of the western tropical South Pacific during austral summer conditions, (2) To study the production and fate of organic matter (including carbon export) on three contrasting trophic regimes (increasing oligotrophy) with a particular emphasis on the role of dinitrogen fixation, and (3) to obtain a representation of the main biogeochemical fluxes and dynamics of the planktonic trophic network. The international OUTPACE cruise took place between 18 February and 3 April 2015 aboard the RV *L'Atalante* and involved 60 scientists (30 onboard). The west-east transect covered ~4 000 km from the western part of the Melanesian Archipelago (New Caledonia) to the western boundary of the South Pacific gyre (French Polynesia). Following an adaptive strategy, the transect initially designed along the $19^o$ S parallel was adapted along-route to incorporate information coming from satellite measurements of sea surface temperature, chlorophyll a concentration, currents, and diazotrophs quantification. After providing a general context and describing previous work done in this area, this introductory paper elucidates the objectives of OUTPACE, the implementation plan of the cruise, water mass and climatological characteristics, and concludes with a general overview of the other papers that will be published in this special issue.

## 1 General context

The additional carbon dioxide ($CO_2$) in the atmosphere, mainly resulting from fossil fuel emissions linked with human activities (anthropogenic $CO_2$), is the main cause of global warming (Fifth Assessment Report - Climate Change 2013 - IPCC). The ocean has acted thus far as a major sink of anthropogenic $CO_2$ (Sabine et al., 2004), preventing greater $CO_2$ accumulation in the atmosphere and therefore greater increase of the Earth's temperature. The biological pump (Fig. 1), the

process by which carbon (C) is transferred from the upper to the deep ocean by biological processes, provides the main explanation for the vertical gradient of C in the ocean. Its strength and efficiency depends upon the complex balance between organic matter production in the photic zone and its remineralisation in both the epipelagic and mesopelagic zones. Before present and for tens of thousands years, the biological pump was thought to be in an equilibrium state with an associated near-zero net exchange of $CO_2$ with the atmosphere (Broecker, 1991; Murnane et al., 1999). Climate alterations then started to disrupt this equilibrium and the expected modification of the biological pump will probably considerably influence oceanic C sequestration (and therefore global warming) in future decades (Sarmiento and Gruber, 2006). The long term decrease of phosphate availability, and the shift from previously nitrogen (N) to phosphorus (P) limited primary production associated with increasing inputs of N by dinitrogen ($N_2$) fixation observed at the Hawaii Ocean Time Series (HOT) station in the north Pacific gyre (NPG) (Karl et al., 1997; Karl, 2014), appear as a first example of biological pump alteration.

The input of new N to the surface ocean through biological $N_2$ fixation represents a major link between the C and N biogeochemical cycles, relating upper ocean nutrient availability with the biological pump, and ultimately the ocean and climate. This link was recently shown to play a central role in previous natural climate changes over long time scales (Galbraith et al., 2013). It is nevertheless clear that expected climate change due to anthropogenic atmospheric $CO_2$ input may concern shorter time scales: the increase of atmospheric $CO_2$ over the past 200 years equals the increase of atmospheric $CO_2$ between glacial to interglacial periods, which took place over several thousands of years (Sarmiento and Gruber, 2006). Furthermore, enhanced stratification in the tropical and subtropical ocean resulting from global warming (Polovina et al., 2008) might decrease nutrient availability such as nitrate ($NO_3^-$), potentially favouring $N_2$ fixation in surface waters. It may also decrease phosphate availability (Moutin et al., 2008), and in turn $N_2$ fixation, net primary production and export (Karl, 2014). McMahon et al. (2015) covering the past 1000 years, argue that $N_2$ fixation has increased since the industrial revolution, and might provide a negative feedback to rising $CO_2$.

The ecosystem changes due to climate change are complex and it is therefore necessary to characterize in detail the interactions between $N_2$ fixation and the C cycle to obtain a precise representation of the $N_2$ fixation process in global biogeochemical models, leading eventually to predictions. Even if considerable scientific progress has been made over the last decades (see reviews from (Sohm et al., 2011; Zehr and Kudela, 2011), many questions remain regarding the impact of this process on biogeochemical cycles and C export.

## 2 The role of $N_2$ fixation in the oligotrophic ocean and an overview of previous cruises in the western tropical South Pacific Ocean (WTSP)

The efficiency of oceanic C sequestration depends upon many factors, among which is the availability of nutrients to support phytoplankton growth in the photic zone (Fig. 1). Large amounts of N are required for phytoplankton growth, as it is an essential component of proteins, nucleic acids and other cellular constituents. Fixed N in the form of $NO_3^-$ or ammonium ($NH_4^+$) is directly usable for growth, but concentrations are low (<0.1 µmol $L^{-1}$) and often growth limiting in most of the

open ocean euphotic zone (Falkowski et al., 1998; Moore et al., 2013). Dissolved $N_2$ gas concentrations in seawater are in contrast very high in the euphotic zone (ca. 450 µmol $L^{-1}$) and could constitute a nearly inexhaustible N source for the marine biota. However, only some prokaryotic organisms (Bacteria, Cyanobacteria, Archaea) hereafter referred to as '$N_2$-fixing organisms' (or diazotrophs) are able to use this gaseous N source since they possess the nitrogenase enzyme that breaks the triple bond between the two N atoms of the $N_2$ molecule, and converts it into a usable form (i.e. $NH_3$) (Zehr et al., 1998). At the global scale, they provide $140 \pm 50$ Tg N per year to the surface ocean, contributing more than atmospheric and riverine N inputs (Gruber, 2004). $N_2$-fixing organisms thus act as 'natural fertilizers', and contribute to sustain life and potential C export in coastal and oceanic environments.

Most of the surface ocean (60 %, Longhurst, 1998) is comprised of low-nutrient, low-biomass oligotrophic areas, which constitute the largest coherent ecosystems on our planet. They support a large part (40 %) of the photosynthetic C fixation in the ocean (Antoine et al., 1996). This C fixation is mainly performed by picoplankton (smaller than 2-3 µm in diameter) that are generally thought to represent a negligible fraction of the total particulate organic C (POC) export flux due to their small size, slow individual sinking rates, and tight grazer control that leads to high rates of recycling in the euphotic zone. Consequently, the efficiency of the biological C pump in these oligotrophic systems has long been considered to be low as the greatest proportion of fixed C is thought to be recycled in the surface layer and rapidly re-exchanged with the atmosphere.

Recent studies have challenged this view and indicate that all primary producers, including picoplanktonic cells, contribute to export from the surface layer of the ocean at rates proportional to their production rates (Richardson and Jackson, 2007). Export mechanisms differ compared to larger cells as export of picoplankton is mainly due to packaging into larger particles via grazing and/or aggregation processes (Jackson, 2001; Lomas et al., 2010). More recently, Close et al. (2013) pointed out that 40-70 % of picoplanktonic cells are small enough to escape detection under the most common definition of suspended particulate organic matter (POM).

Besides submicron POM export, low $\delta^{15}N$ signatures in particles from sediment traps at the HOT station suggest that at least part of the production sustained by $N_2$ fixation is ultimately exported out of the photic zone (Karl et al., 2012; Karl et al., 1997; Scharek et al., 1999a; Sharek et al., 1999b). This may either be direct through sinking of diazotrophs, or indirect, through the transfer of diazotroph-derived N to non-diazotrophic plankton in the photic zone, that is subsequently exported. Karl et al. (2012) reported an efficient summer export flux of C (three times greater than the mean wintertime particle flux) at the HOT station that was attributed to the direct export of symbiotic $N_2$-fixing cyanobacteria associated with diatoms (hereafter referred to as diatom-diazotroph associations or DDAs), which have high sinking and low remineralization rates during downward transit. This result is in accordance with high export fluxes measured in the tropical North Atlantic when the diazotroph community is dominated by DDAs (Subramaniam et al., 2008; White et al., 2012). More recently, (Berthelot et al., 2015; Bonnet et al., 2016a) studied the fate of a bloom of unicellular diazotrophs from Group C (UCYN-C) during a

mesocosm experiment in the New Caledonia lagoon after a phosphate enrichment and revealed that ~10 % of UCYN-C from the water column was exported to the particle traps daily, representing as much as 22.4±5.5 % of the total POC exported at the height of the UCYN-C bloom. A $\delta^{15}$N budget performed in the mesocosms confirmed the high contribution of N$_2$ fixation (56 %, Knapp et al., 2016) to export compared to other tropical and subtropical regions where active N$_2$ fixation

contributes 10 to 25 % to export production (e.g. Altabet, 1988; Knapp et al., 2005) and exceptionally up to 92% in the Arabian Sea (Gandhi et al. 2011; Kumar et al. 2017). Mechanistically, the vertical downward flux was enabled by the aggregation of the small (5.7±0.8 µm) UCYN-C cells into large (100-500 µm) aggregates. In addition to direct export of diazotrophs, the use of nanoSIMS (nanoscale Secondary Ion Mass Spectrometry) enabled tracking the fate of $^{15}$N from both Trichodesmium (Bonnet et al., 2016b) and UCYN blooms (Berthelot et al., 2015; Bonnet et al., 2016c), and demonstrated

that ~8% of N originating from N$_2$ fixation is quickly transferred to non-diazotrophic plankton, in particular diatoms (i.e. efficient C exporters to depth, (Nelson et al., 1995) during Trichodesmium blooms (Bonnet et al., 2016b). This reveals that N$_2$ fixation can fuel large-size non-diazotrophic plankton growth in the water column, suggesting an indirect export pathway of organic matter sustained by N$_2$ fixation in the oligotrophic ocean. Most of the above-mentioned studies were performed in microcosms and mesocosms and further open ocean studies combining the set of complementary approaches described

above are needed to better assess the fate of N$_2$ fixation and its role on C export.

The western tropical south Pacific (WTSP) is an ideal location to study the fate of N fixed by N$_2$ fixation, as it is considered a hot spot of N$_2$ fixation in the world ocean (Bonnet et al., 2017a). While average N$_2$ fixation rates range from 20-200 µmol N m$^{-2}$d$^{-1}$ in the tropical North Atlantic (Benavides and Voss, 2015) and Pacific (Böttjer et al., 2017; Dore et al., 2002), they reach 30-5400 µmol N m$^{-2}$d$^{-1}$ (average ~800 µmol N m$^{-2}$d$^{-1}$) in the Solomon Sea (western part of the WTSP) (Berthelot et

al., Submitted; Bonnet et al., 2009; Bonnet et al., 2015), which is in the upper range of rates reported in the global N$_2$ fixation MAREDAT database and even surpassed its upper rates (100-1000 µmol N m$^{-2}$ d$^{-1}$) (Luo et al., 2012). Very high rates have also been recently reported in the Arabian Sea (Gandhi et al. 2011; Kumar et al. 2017). High rates ranging from 151 to 703 µmol N m$^{-2}$ d$^{-1}$ have also been reported off New Caledonia (Garcia et al., 2007), with seasonal variations closely linked with phosphate availability (Moutin et al., 2005; Van Den Broeck et al., 2004). The seasonal distribution of N$_2$

fixation is corroborated by *in situ* and satellite observations (Dupouy et al., 2011) of recurrent large Trichodesmium blooms during austral summer conditions (January-March) over the 1998-2010 period in the Melanesian archipelago (MA: New Caledonia, Vanuatu, Fiji Islands). In addition to Trichodesmium which dominates the diazotroph community in the WTSP (Moutin et al., 2005; Berthelot et al., Submitted; Bonnet et al., 2015), very high abundances of UCYN-B (up to 106-107 nifH copies L$^{-1}$) have been reported (Bonnet et al., 2015; Campbell et al., 2005; Moisander et al., 2010). The uncultivated UCYN-

A (Zehr et al., 2008) also displays high abundances (105-106 nifH copies L$^{-1}$, (Bonnet et al., 2015; Moisander et al., 2010) around the MA, but they seem to have different ecological niches compared to Trichodesmium and UCYN-B (Bonnet et al., 2015; Moisander et al., 2010).

When going eastward towards the South Pacific gyre (GY), Halm et al. (2012) have reported rates of 12-190 µmol N m$^{-2}$ d$^{-1}$ on the western border of the gyre and Raimbault and Garcia (2008) and Moutin et al. (2008) reported rates of 60 ± 30 µmol N m$^{-2}$ d$^{-1}$ in the central gyre, indicating a decreasing gradient of N$_2$ fixation from west to east and low N$_2$ fixation rates relative to other ocean gyre ecosystems. The organisms responsible for these fluxes are different from common autotrophic diazotrophs such as Trichodesmium or UCYN-B, and are mainly affiliated with heterotrophic proteobacteria and low abundances of UCYN-A (Bonnet et al., 2008; Halm et al., 2012).

The west to east zonal gradient of N$_2$ fixation and the distinct diversity of N$_2$-fixing organisms along this gradient together provide a unique opportunity to study how production, mineralisation and export of organic matter depends upon N$_2$ fixation in contrasting oligotrophic regimes (from oligotrophy to ultra-oligotrophy). Comparisons between different systems along a zonal gradient of trophic status and N$_2$ fixation will provide new insights for identifying and understanding fundamental interactions between marine biogeochemical C, N, P, silica (Si), and iron (Fe) cycles in oligotrophic ecosystems.

## 3 Objectives of OUTPACE

The overall goal of OUTPACE was to obtain a precise representation of the complex interactions between planktonic organisms and the cycle of biogenic elements (C, N, P, Si, Fe), considering a variety of scales, from single-cell processes to the whole WTSP Ocean. To meet this goal, the three specific objectives of OUTPACE were the following:

1) To perform a zonal characterization of the biogeochemistry and biological diversity of the WTSP during the strongest stratified period (austral summer),

2) To study the production and fate of organic matter (including C export) of three contrasting environments (from oligotrophy to ultra-oligotrophy) with a particular emphasis on N$_2$ fixation,

3) To obtain a representation of the main biogeochemical fluxes (C, N, P, Si, Fe) and the dynamics of the planktonic trophic network, both *in situ* and by using microcosm experiments.

The detailed study of biological production and its subsequent fate at a given site implied a combination of adaptive and Lagrangian strategies. Indeed, as pointed out by d'Ovidio et al. (2015), the spatiotemporal domain of an oceanographic cruise is also one in which horizontal stirring generated by ocean circulation at the mesoscale induces strong variability in biogeochemical tracers' distributions. Consequently, ephemeral and local gradients due to mesoscale activity can easily mask the large-scale gradients. Following d'Ovidio et al. (2015), this problem can be overcome by adopting an adaptive sampling strategy (described in Sect. 4.1.) based on information on sea surface temperature (SST), chlorophyll a (chl a) concentrations and currents provided by satellite products analyzed in real time during the entire cruise.

**4 Implementation of the OUTPACE cruise**

The OUTPACE cruise was conducted during austral summer conditions from 18 February to 3 April 2015, aboard the RV *L'Atalante*. We performed a ~4,000 km zonal transect from the North of New Caledonia to the western part of the GY, finally reaching French Polynesia (Fig. 2). Along this transect, two types of stations were sampled: 15 SD "short duration" (8
5    h) stations dedicated to a large-scale description of biogeochemical and biodiversity gradients; and three LD "long duration" (7 days) stations for Lagrangian process studies.

**4.1 Adaptive strategy**

Following the planned adaptive strategy, the initial transect designed to approximatively follow 19° S was modified along-
route thanks to the information coming from satellite images. The regions along the vessel route were first characterized at large scale through the analysis of satellite (Altimetry, SST, Ocean color) data. These data were automatically retrieved and processed to derive Eulerian and Lagrangian diagnostics of ocean circulation and biogechemistry: Okubo-Weiss parameter maps, Lagrangian Coherent Structures (LCS), and chl a maps (d'Ovidio et al., 2015).

The satellite data was treated in near real time on land and the obtained data were transmitted on board together with a daily
bulletin containing the analysis of remote sensing information, along with suggestions for LD station positions (the complete series of the 42 bulletins is available on the OUTPACE website at https://outpace.mio.univ-amu.fr, section "Adaptive Strategy").

Two main criteria were adopted in suggesting LD station positions:

1) The areas for the LD A and LD B (LD C) stations were characterized by local maxima (minima) of sea surface chl a
concentration to sample MA (GY) conditions, and

2) Local minima of surface current intensity for all LD stations, to increase the chance of sampling a homogeneous water mass.

Once the suggested positions for LD stations were relayed via the daily bulletin, at one of these locations real time analysis of water samples by quantitative Polymerase Chain Reaction (qPCR) was conducted to measure abundances of six groups of
diazotrophs (Stenengren et al., this issue). In this way, we located regions with diazotrophs, while also resolving the contrasting role Trichodesmium spp. or UCYN-dominated diazotroph communities have on biogeochemical cycles.

Finally, the exact locations of the three LD stations were then determined on board in real time from a rapid survey using a Moving Vessel Profiler (MVP) equipped with conductivity-temperature-depth (CTD) and fluorimeter sensors, accompanied by the hull-mounted thermosalinograph and acoustic Doppler current profiler.
During this rapid survey, it was planned to follow two different sampling routes: a cross of about 40 km each side, followed by a Zig-Zag route covering an area of 25 km each side at the centre of the cross. This strategy was applied as planned for the LD A and LD C stations.

The LD A station was performed east of the northern extremity of New Caledonia in an anticyclonic recirculation characterized by a relative high surface chl a concentration. The LD C station was performed in a cyclonic eddy in the most oligotrophic part of the OUTPACE transect (GY) close to Cook Islands.

The severe meteorological conditions due to the development of tropical cyclone Pam (a category 5 storm) that hit the Vanuatu Islands, obliged us to perform the LD B station at a more easterly location than initially planned. Satellite imagery allowed for the targeting of a large filament of high surface chl a concentration close to Niue Island. Due to the circulation patterns associated with the bloom, the rapid survey strategy was adapted in order to perform four sections across the structure (see details in de Verneil et al., this issue).

SD station positions were chosen in relation to the LD stations, so that they were roughly equidistant from each other, respected territorial waters, and incorporated the changing conditions during the cruise.

## 4.2 General programme at each station

Every station (Table 1) was investigated from the surface to 2000 m.

SD stations

Each of the 15 SD stations was investigated for 8 h. Specific operations during the SD station occupation consisted of:

(1) Two 0-200 m CTD casts and Niskin bottle sampling and one 200-2000 m CTD cast and Niskin bottle sampling using the classical SBE 9+ CTD-Rosette (C-R) for measurements of core parameters (dissolved oxygen, dissolved inorganic carbon, total alkalinity, nutrients, chl a, Particulate and dissolved organic C, N, P and Si) and some more specific ones (for example primary production rates, $N_2$ fixation rates and diazotroph abundance). An Underwater Vision Profiler (UVP) was attached to the CTD-rosette to quantify and visualize suspended particulate material;

(2) One 0-500 m CTD cast and bottle sampling using the Trace Metal Clean SBE 9+ CTD-Rosette (TM-R) equipped with 24 teflon-lined GoFlo bottles to sample for trace metals analyses;

(3) Optical sensors casts: Integrated measurements of bio-optical properties and pigments were made with instruments measuring hyperspectral radiometry in the UV-Visible domain with UV-VIS Trios spectroradiometers, and a MicroPro free-fall profiler (Satlantic) was used for downward irradiance measurments.

(4) Hauls for phytoplankton and zooplankton sampling with specific nets; and

(5) A profile of turbulence measurements using a VMP1000 equipped with microsensors for temperature and shear that enable accurate estimates of the eddy diffusion coefficient Kz.

The specific configurations of the two CTD-Rosettes are available here: https://outpace.mio.univ-amu.fr/spip.php?article137.

LD stations

Each of the three LD stations was investigated with a drifting array (see below) that was deployed for 7 days. A series of CTD (C-R) casts (0-500 m) were performed every 3 h near the actual position of the drifting array while numerous specific operations (see below) were carried out in between CTD casts.

A total of 13 Surface Velocity Program (SVP) drifters anchored at 15 m depth were deployed in order to study relative surface dispersion. The drifters were launched with three at LD A, six at LD B and four at LD C. A drifting array (equipped with three PPS5 sediment traps, current meters, specific oxygen sensors, and specific high frequency temperature sensors; please consult https://outpace.mio.univ-amu.fr/spip.php?article75 for details) was then deployed at the chosen station position to start the process study. The drifting array was recovered at the end of each LD station occupation, immediately following the last CTD cast. An additional CTD cast from surface to bottom (5000 m) was undertaken at the LD B station.

Specific operations during LD stations were identical to those performed at SD stations, along with other operations like *in situ* production measurements, *in situ* particulate material sedimentation measurements, trace metal clean pumping for process experiments, and profiles of current measurements. Finally, at each LD station, a drifting Provor-type ARGO float (ProBio equipped with sensors to measure chl a (fluorescence), CDOM (fluorescence), PAR, Irradiance at 3 wavelengths, backscattering and dissolved oxygen (optode)), was deployed (Table 2).

The following color code was proposed to present data from the different LD stations: LD A green; LD B red; and LD C blue.

All details regarding the OUTPACE cruise are available on the OUTPACE web site: https://outpace.mio.univ-amu.fr/.

The general scheduled and realized programmes are available here: http://www.obs-vlfr.fr/proof/php/outpace/outpace_log_and_basic_files.php

## 5 General characteristics of the upper water masses in the WTSP

Water characteristics (temperature, salinity, density, chl a fluorescence) for the upper 700 m as measured by the C-R are presented in Fig. 3. The deep CTD casts from all SD and LD stations are presented subsequent to post-cruise processing using Sea-Bird Seasoft software adopting the TEOS-10 standard. The upper surface layer (0-30 m) observed during the OUTPACE transect was characterized by warm waters, with temperatures between 26.18 and 29.93 °C (Fig. 3a), and relatively low salinity, i.e. absolute salinity between 35.03 and 35.81 g kg$^{-1}$ (Fig. 3b). Density anomalies of 21.72-22.91 kg.m$^{-3}$ between 0-30 m increased gradually between 30 and 200 m to reach 24.89-25.38 kg m$^{-3}$ (Fig. 3c). The salinity increase of subsurface waters (100-200 m) from 187°W longitude (Fig. 3b) indicated the geographical border between waters under the influence of the MA (SD1 to SD12) and waters from the GY (SD13 to SD15, LD C). LD B was not classified here and required a further analysis (de Verneil et al., this issue). The classification between MA and GY waters will be helpful to describe general biogeochemical and biological features. Between 200 and 700 m, a decreasing gradient of temperature and salinity indicated the presence of permanent thermocline waters. Temperature fell from 19.27-22.08 °C at 200 m to 5.48-6.91 °C at 700 m (Fig. 3a). Absolute salinities of 35.76-36.21 g kg$^{-1}$ at 200 m decreased to 34.49-34.61 g kg$^{-1}$

(Fig. 3b). The density anomalies of 24.89-25.38 kg m$^{-3}$ at 200 m increased largely to 26.97-27.13 kg m$^{-3}$ at 700 m (Fig. 3c). The maximum fluorescence depth, considered here as the main indicator of the trophic state, increased from ~100 m depth in the MA waters to ~115-150 m depth in the GY waters, which allowed us to sample the oligotrophic to ultraoligotrophic transition in the WTSP for the purpose of the OUTPACE project.

**6 The climatological context of the campaign**

The OUTPACE cruise took place in the WTSP, a region impacted by the El-Nino Southern Oscillation (ENSO), known to be the most important mode of SST variability on inter-annual to decadal time scales (Sarmiento and Gruber, 2006). ENSO-related SST anomalies are caused by a combination of changes in ocean circulation (mainly changes in the strength and source of equatorial upwelling) and anomalous local air-sea heat exchanges. The most dramatic effects of ENSO in the

surface ocean are well documented in the Eastern Tropic Pacific, where seasonal upwelling conditions can be suppressed with severe economic consequences for fisheries (Chavez et al., 2003). During El-Niño phases (negative Southern Oscillation Index (SOI), defined below), the warm pool normally positioned in the Western Pacific is found farther east, resulting in the aforementioned suppression of upwelling conditions off Peru. La Niña, the opposite phase with positive SOI, reverses this situation. After decades of intense study, ENSO is still an active field of research (Takahashi et al., 2011).

Given the known importance of ENSO for the Tropical South Pacific, it is worthwhile determining in which climatological conditions the cruise was performed. To achieve this goal, we identified years of opposing ENSO phase and analyzed the corresponding WTSP conditions with available satellite data. ENSO phases were identified using the monthly time series of Southern Oscillation Index (SOI) provided by NCEP (http://www.cpc.ncep.noaa.gov/, downloaded 5 December 2016). The SOI metric uses differences in standardized sea level air pressure between Papeete (Tahiti) and Darwin (Australia) to

represent ENSO phase as previously mentioned. The OUTPACE region lies between these two locations (Fig. 2), highlighting again ENSO's possible influence on the cruise.

WTSP conditions were estimated with SST and surface chl a concentration measured by the MODIS Aqua satellite mission and available at the NASA Ocean Color Data website (https://oceandata.sci.gsfc.noaagov/, downloaded 14 December 2016). Global annual and monthly (March) averages of both SST and chl a were provided by NASA at level 3 (i.e. mapped) with 4

25    km satellite pixel resolution, resulting in four separate datasets from 2003 to present. The data within the OUTPACE region, defined between 25° S, 155° E, and 15° S, 149° W to envelope the cruise track as in Fig. 2, were extracted. March was chosen as the month of study because it was the central month of the cruise. Plots of these four datasets can be found on the OUTPACE dataset website, section "http://www.obs-vlfr.fr/proof/php/outpace/outpace_figures.php". Probability density distributions were generated for each dataset in its entirety. Additional probability distributions were also calculated on data

subsets, for years 2003 and 2011 as chosen by SOI to represent opposite ENSO phases, along with 2015, the year of the cruise. In order to gauge significance between probability distributions of different years, for each of the four datasets the

temporal standard deviation of each pixel was calculated. This resulted in a distribution of standard deviations, the median of which was taken to represent inherent inter-annual variability.

The time series of SOI is presented in Fig. 4, with El Niño (La Niña) values shaded red (blue). During OUTPACE cruise sampling in austral summer 2015, a strong El Niño was observed. The other El Niño year considered here was 2003, chosen because of its relative strength and duration similar to 2015; for similar reasons of intensity and duration, 2011 was designated a representative La Niña year.

The probability density distributions for the annual and March means of both SST and chl a are presented in Fig. 5. In SST, the March mean showed a larger proportion of warmer temperatures in relation to the annual mean (white lines in Fig. 5a,c), reflecting the austral summer season. The 2011 La Niña distributions showed a greater proportion of warm temperatures in both annual and March distributions (blue lines in Fig. 5a,c), which is consistent with the idea of La Niña accentuating the warm pool present in the WTSP. The probability peak of March 2011 SST was much more localized, between 26 and 30° C, than 2003, 2015, and the mean (Fig. 5c). The 2003 March SST (red line in Fig. 5c) was the widest distribution, with a slight rightward skew. By contrast, both the mean and 2015 March probabilities (white and green lines, respectively) had a left skew. Considering the median 95% confidence interval for the March SST time series, the central peaks for these distributions could not be considered distinct.

Satellite chl a showed distributions completely different to SST. For all quantities considered, the annual and March mean (Fig. 5b,d), as well as the years 2003, 2011, and 2015, showed a bi-modal distribution, where the two peaks were distinct enough from each other to be significant from the median pixel variability. Interestingly, 2003 and 2011 (red and blue lines, Fig. 5b) annual chl a distributions overlapped considerably more than 2015 (green line), which had its entire probability distribution shifted to the right, though this shift may not have been significant. The March mean for 2003 and 2015 chl a (red and green lines, respectively, Fig. 5d), however, almost entirely overlapped, possibly signifying that El Niño chl a distributions were more alike than La Niña years. The annual 2015 mean of chl a being slightly different from other years, and yet the March 2015 mean resembling 2003, might indicate that 2015 was indeed a slightly different year from a surface chlorophyll perspective, but this slight difference was concentrated in other parts of the year than late austral summer when OUTPACE took place.

In summary, the SOI metric of ENSO showed that 2015 was an El Niño event, and both the SST and chl a satellite data in the WTSP partially reflected this. The March 2003 and 2015 SST distributions resembled each other, but they also resembled the entire dataset more than the 2011 La Niña distribution. From the SST point of view, perhaps La Niña events impact the region more than El Niño. Chl a distributions for the El Niño years also overlapped more than with La Niña, but the variability inherent the time series precluded declaring significant differences. Overall, in the WTSP both SST and chl a were not atypical from what one would expect, and so we determined that climatological effects upon the results of OUTPACE were minimized.

# 7 Special issue presentation

The goal of this special issue is to present the knowledge obtained concerning the functioning of WTSP ecosystems and associated biogeochemical cycles based on the datasets acquired during the OUTPACE experiment. The cruise strategy was organized to promote collaboration between physicists, biologists and biogeochemists with expertise including marine physics, chemistry, optics, biogeochemistry, microbiology, molecular ecology, genetics, and modelling. Most of the contributions to this volume have benefited from this collective effort and are presented according to the main objectives of the OUTPACE experiment.

The hydrological and dynamical context of biogeochemical sampling are described for the entire cruise route (Fumenia et al., 2017) and specifically at the three long duration stations, where low physical variability validated the quasi-lagrangian sampling strategy employed (de Verneil et al., 2017a). Turbulence measurements revealed an interesting longitudinal gradient with higher turbulence levels in the West, i.e. the Coral Sea, compared to the Eastern part within the gyre, consistent with the increasing oligotrophy (Bouruet-Aubertot et al., 2017). The large scale circulation was dominant even though the mesoscale and submesoscale circulation can have a strong influence (Rousselet et al., 2017), in particular on the bloom observed at station LD B (de Verneil et al., 2017b).

An important focus of OUTPACE was on dinitrogen fixation and its fate in the ecosystem (Caffin et al., 2017b). $N_2$ fixation was detected at all 18 sampled stations and the transect could be divided into two main characteristic sub-areas (Bonnet et al., 2017): i) the MA waters (160°W to 170°E) exhibiting very high $N_2$ fixation rates ($631 \pm 286$ µmol N m$^{-2}$ d$^{-1}$, i.e. among the highest reported for the global ocean, Luo et al., 2012) and dominated by Trichodesmuim (Stenengren et al., 2017), and ii) the GY waters (170°E-160°E) exhibiting low $N_2$ fixation rates (average $85 \pm 79$ µmol N m$^{-2}$ d$^{-1}$) dominated by UCYN-B (Stenengren et al., 2017). The differing $\delta^{15}N$ signature of suspended particles measured over the photic layer of MA (-0.41‰) and GY waters (8.06‰) confirms the presence of two contrasting regions in terms of $N_2$ fixation. Thanks to the lagrangian strategy followed at the LD stations, and the low dispersion measured showing that we sampled the same water masses (de Verneil et al., 2017a), N-budgets were established (Caffin et al. 2017a). $N_2$ fixation was the major external source of N representing more than 90 % of new N input into the photic layer, and the e-ratio quantifying the efficiency of a system to export particulate organic matter was higher in MA waters than in GY waters (Caffin et al., 2017a). Caffin et al. (2017b) revealed that the diazotroph-derived nitrogen (DDN) was efficiently transferred from diazotrophs (Trichodesmium and UCYN) to non-diazotrophic phytoplankton, both autotrophs and heterotrophs. Hunt et al., (2017) report an efficient transfer of DDN in zooplankton. The fate of C and N was under the influence of programmed cell death in diazotrophs (Berman-Frank et al., 2017) but diazotrophs were poorly exported directly, and we suspect that this transfer of DDN fueled indirect export associated with $N_2$ fixation. By using nitrogen isotope budgets, Knapp et al., (2017) confirmed that >50% of export production was supported by $N_2$ fixation in MA waters. Stenegren et al. (2017) identified a clear niche separation between a subsurface (UCYN-A1 and A2 with their hosts) and a surface group (Trichodesmium, UCYN-B and the het-group) based on a temperature-depth gradient. They also found discrepancies between the UCYN-A and their hosts in both abundances and

distributions which suggest that the UCYN-A could be living freely or with a wider diversity of hosts than previously believed. Finally, the assemblage of epibiotic micro-organisms associated with Trichodesmium were characterized in relation with environmental parameters (Frischkorn et al., 2017). $N_2$ fixation in the ocean does not only occur in tropical sunlit surface waters, but also in less obvious environments such as temperate latitudes and aphotic waters. Here, $N_2$ fixation

was measured also in the mesopelagic zone along the OUTPACE transect and the diazotroph community present were identified. Deep $N_2$ fixation rates were low but measurable and recurrently found along the transect with the exception of the easternmost stations located in the ultraoligotrophic subtropical Pacific gyre (Benavides et al., 2017). $N_2$ fixation activity was presumably driven by the dominating Gamma-proteobacterial community, and fueled by the presence of labile organic matter compounds. Benavides et al. (2017) provided further evidence that $N_2$ fixation in the deep ocean is not negligible and

likely impacts global nitrogen inputs in a significant manner.

The dynamics of phytoplankton (Bock et al., 2017; Leblanc et al., 2017; Lefevre et al., 2017; Guidi, 2017), heterotrophic bacterioplankton (Van Wambeke et al., 2017), and zooplankton (Carlotti et al., 2017; Hunt et al., 2017) along the zonal gradient of diazotroph diversity and activity are described together with the composition and distribution of dissolved organic carbon (Panagiotopoulos et al., 2017) and the changes in inorganic carbon content along the longitudinal transect

(Wagener et al., 2017). Stations sampled during the OUTPACE cruise were characterized by a highly stratified community structure, with significant contributions of Prochlorococcus and picophytoeukaryote populations to biomass (Bock et al., 2017). Size-fractionated results show a non-negligible contribution of the pico-sized fraction (<2-3 µm) to both Si biomass and uptake, which could confirm the previous hypothesis of Si assimilation by Synechoccocus populations or reflect the presence of an overlooked Si group such as Parmales (Leblanc et al., 2017). Surface DOC concentrations varied little (50-75

µM) across the transect with slightly higher values observed at LD B (78 µM), and labile organic matter (sugars were used as a good proxy) closely followed DOC patterns ranging from 1.5 to 3 µM with higher values also recorded at LDB (3.5 µM). Labile organic matter accounted about 3-5% of DOC with glucose being the dominant sugar (> 60% of total sugars) (Panagiotopoulos et al., 2017). Valdes et al., (2017) suggest that copepods can retain N and P compounds obtained from feeding in the upper layer, preventing the rapid loss of these nutrients. Copepods were able to sustain and modify the

composition of microbial communities and could provide P for further development of cyanobacterial blooms.

Optical properties of the WTSP waters are presented, with a focus on the cyanobacterial (diazotroph) impact upon bio-optical properties, UV-VIS light attenuation (Dupouy et al., 2017) and chl a algorithms (Frouin et al., 2017). Operational NASA bio-optical algorithms (OC4v6, OCI) substantially underestimated surface chl a concentration, but a normalized reflectance difference index, robust to atmospheric correction errors, performed well over the range of chl a values

encountered across the transect (Frouin et al., 2017). Trichodesmium is considered the main nitrogen-fixing species, especially in the South Pacific region. Due to the paucity of in situ observations, alternative methods for estimating the presence of Trichodesmium must be sought to evaluate the global impact of these species on primary production. Rousset et al. (2017) elaborate a new satellite-based algorithm and use that algorithm to estimate the extent of Trichodesmium surface blooms and their dynamics during the OUTPACE experiment. Finally the main processes controlling the biological carbon

pump in the WTSP were investigated using 1DV (Gimenez et al., 2017) and regional (Dutheil et al., 2017) biogeochemical-physical coupled models. The new knowledge gained on the interactions between planktonic organisms and the cycle of biogenic elements are then used to propose a new scheme for the biological carbon pump functioning and its role, at the present time and in the near future, in the oligotrophic Pacific Ocean (Moutin et al., 2017).

**Acknowledgements**

This is a contribution of the OUTPACE (Oligotrophy from Ultra-oligoTrophy PACific Experiment) project (https://outpace.mio.univ-amu.fr/) funded by the French research national agency (ANR-14-CE01-0007-01), the LEFE-CyBER program (CNRS-INSU), the GOPS program (IRD) and the CNES (BC T23, ZBC 4500048836). The OUTPACE

cruise (http://dx.doi.org/10.17600/15000900) was managed by MIO (OSU Institut Pytheas, AMU) from Marseilles (France). The authors thank the crew of the RV *L'Atalante* for outstanding shipboard operations. G. Rougier and M. Picheral are warmly thanked for their efficient help in CTD rosette management and data processing, as well as C. Schmechtig for the LEFE-CyBER database management. The satellite-derived data of Sea Surface Temperature, chl a concentrations and currents have been provided by CLS in the framework of the CNES funding; we warmly thank I. Pujol and G. Taburet for

their support in providing these data. We acknowledge NOAA, and in particular R. Lumpkin, for providing the SVP drifters. All data and metadata are available at the following web address: http://www.obs-vlfr.fr/proof/php/outpace/outpace.php

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

Table 1. Date, location and general characteristics of the stations investigated along the OUTPACE transect. Distance in km from the first SD station (SD1).

| Station | | CTD cast (0-2000 m) | Arrival date (UTC) | Departure date (UTC) | Latitude (deg min) | | | Longitude (deg min) | | | Latitude (deg) | Longitude (deg) | Cumulative distance (km) | Bottom Depth (m) |
|---|---|---|---|---|---|---|---|---|---|---|---|---|---|---|
| Nouméa | | | x | 19/02/2015 21:00 | 22 | 14 | | 166 | 28 | | -22,23 | 166,47 | x | 0 |
| SD | 1 | 007 | 21/02/2015 20:00 | 22/02/2015 09:30 | 18 | 0 | S | 159 | 54 | E | -18,00 | 159,90 | 0 | 4068 |
| SD | 2 | 016 | 22/02/2015 21:45 | 23/02/2015 10:00 | 18 | 38 | S | 162 | 8 | E | -18,63 | 162,12 | 210 | 2567 |
| SD | 3 | 020 | 24/02/2015 03:45 | 24/02/2015 10:00 | 19 | 0 | S | 164 | 54 | E | -19,00 | 164,90 | 488 | 3252 |
| LD | A | 067 | 25/02/2015 13:00 | 02/03/2015 23:15 | 19 | 13 | S | 164 | 41 | E | -19,21 | 164,69 | 528 | 3491 |
| SD | 4 | 071 | 04/03/2015 08:30 | 04/03/2015 15:15 | 20 | 0 | S | 168 | 0 | E | -20,00 | 168,00 | 839 | 4995 |
| SD | 5 | 075 | 05/03/2015 05:45 | 05/03/2015 13:30 | 22 | 0 | S | 170 | 0 | E | -22,00 | 170,00 | 1103 | 4405 |
| SD | 6 | 079 | 06/03/2015 03:15 | 06/03/2015 12:30 | 21 | 22 | S | 172 | 8 | E | -21,37 | 172,13 | 1303 | 2509 |
| SD | 7 | 083 | 07/03/2015 00:15 | 07/03/2015 09:30 | 20 | 44 | S | 174 | 16 | E | -20,73 | 174,27 | 1506 | 2451 |
| SD | 8 | 087 | 07/03/2015 21:00 | 08/03/2015 07:15 | 20 | 42 | S | 176 | 24 | E | -20,70 | 176,40 | 1694 | 2028 |
| SD | 9 | 091 | 08/03/2015 22:15 | 09/03/2015 07:45 | 20 | 57 | S | 178 | 39 | E | -20,95 | 178,65 | 1927 | 3864 |
| SD | 10 | 095 | 10/03/2015 00:00 | 10/03/2015 07:15 | 20 | 28 | S | 178 | 31 | W | -20,47 | 181,48 | 2187 | 819 |
| SD | 11 | 099 | 10/03/2015 21:45 | 11/03/2015 05:15 | 19 | 59 | S | 175 | 40 | W | -19,98 | 184,33 | 2449 | 2234 |
| SD | 12 | 103 | 11/03/2015 21:00 | 12/03/2015 05:00 | 19 | 29 | S | 172 | 50 | W | -19,48 | 187,17 | 2711 | 7717 |
| LD | B | 151 | 15/03/2015 12:00 | 20/03/2015 22:30 | 18 | 14 | S | 170 | 52 | W | -18,24 | 189,14 | 2985 | 4912 |
| SD | 13 | 152 (0-500 m) | 21/03/2015 10:30 | 21/03/2015 11:00 | 18 | 12 | S | 169 | 4 | W | -18,20 | 190,93 | 3096 | 4598 |
| LD | C | 199 | 23/03/2015 12:00 | 28/03/2015 22:00 | 18 | 25 | S | 165 | 56 | W | -18,42 | 194,06 | 3371 | 5277 |
| SD | 14 | 210 | 30/03/2015 01:30 | 30/03/2015 09:15 | 18 | 25 | S | 163 | 0 | W | -18,42 | 197,00 | 3640 | 4998 |
| LD | 15 | 213 | 31/03/2015 00:30 | 31/03/2015 08:30 | 18 | 16 | S | 160 | 0 | W | -18,27 | 200,00 | 3916 | 4965 |
| Papeete | | | 02/04/2015 21:00 | x | 17 | 34 | S | 149 | 36 | W | -17,57 | 210,40 | x | 0 |

Table 2. Provor ARGO floats deployed along the OUTPACE transect.

| Station | Float number WMO id. | Deployment location Latitude | Longitude | Date of deployment | Time (UTC) | closest CTD cast |
|---|---|---|---|---|---|---|
| LD A | ProBio075b | 19° 13.00 S | 164° 29.40 W | 03/03/2015 | 22:45:00 | OUT_C_067 |
| SD 11 | 6901663 | 19° 59.56 S | 175° 38.59 W | 11/03/2015 | 05:03:00 | OUT_C_099 |
| SD 12 | 6901664 | 19° 32.61 S | 172° 46.47 W | 12/03/2015 | 05:00:00 | OUT_C_103 |
| LD B | 6901666 | 19° 44.50 S | 170° 31.31 W | 13/03/2015 | 01:15:00 | |
| LD B | 6901667 | 17° 38.39 S | 170° 59.47 W | 13/03/2015 | 18:49:00 | |
| LD B | 6901668 | 18° 13.90 S | 170° 44.20 W | 20/03/2015 | 21:36:00 | OUT_C_151 |
| LD B | ProBio077b | 18° 16.09 S | 170° 43.80 W | 20/03/2015 | 22:00:00 | OUT_C_151 |
| Before LD C | 6901669 | 18° 46.92 S | 168° 09.06 W | 22/03/2015 | 07:55:00 | |
| LD C | 6901670 | 18° 41.20 S | 165° 45.18 W | 22/03/2015 | 19:48:00 | |
| LD C | 6901671 | 18° 28.16 S | 165° 46.21 W | 28/03/2015 | 21:25:00 | OUT_C_199 |
| LD C | ProBio079b | 18° 28.16 S | 165° 46.21 W | 28/03/2015 | 21:30:00 | OUT_C_151 |
| SD 14 | 6901679 | 18° 24.26 S | 162° 59.34 W | 30/03/2015 | 09:45:00 | OUT_C_210 |
| SD 15 | 6901680 | 18° 15.29 S | 159° 59.23 W | 31/03/2015 | 08:15:00 | OUT_C_213 |

**Figures caption**

Fig. 1. Major C fluxes for a biological pump budget and main role of $N_2$ fixation. Biological pump: C transfer by biological processes into the ocean interior. DIC: Dissolved Inorganic C, POC: Particulate Organic C, DOC: Dissolved Inorganic C. See Moutin et al., (2012) for a detailed description.

Fig. 2. Transect of the OUTPACE cruise superimposed on (a) arithmetic mean surface chl a and (b) quasi-Lagrangian weighted mean chl a of the WTSP during OUTPACE. The two types of station, short duration (X) and long (+) duration investigated for a period longer than seven days, are indicated. The satellite data are weighted in time by each pixel's distance from the ship's average daily position for the entire cruise. The white line shows the vessel route (data from the hull-mounted ADCP positioning system). Coral reefs and coastlines are shown in black, land is grey, and areas of no data are left white. The ocean color satellite products are produced by CLS with support from CNES.

Fig. 3. Zonal sections of (a) conservative temperature Θ, (b) absolute salinity SA, (c) density anomaly, and (d) fluorescence in the upper 700 m from the classic CTD rosette during SD and LD stations along the OUTPACE transect. The three LD stations are highlighted by their color-coded letter and corresponding arrow.

Fig. 4. Time series of monthly Southern Oscillation Index (SOI) from January 2000 to October 2016. Negative and positive values are shaded red and blue to signify El Niño and La Niña, respectively. SOI smoothed by 'lowess' filter with a five point window is shown as a solid black line. Black arrows indicate March for the years where satellite data are available. Dashed vertical lines indicate March 2003, 2011, and 2015.

Fig. 5. Probability density estimates for MODIS Aqua data in the OUTPACE region, using mean (a) Annual SST, (b) Annual chl a, (c) March SST, and (d) March chl a. Probability densities for the ensemble of all years is shown in white, while densities for 2003, 2011, and 2015 are in red, blue, and green, respectively. Also plotted are 95% confidence intervals (two standard deviations) for each subset estimated using the median variance of pixel inter-annual variability. Note the logarithmic scale for chl a.

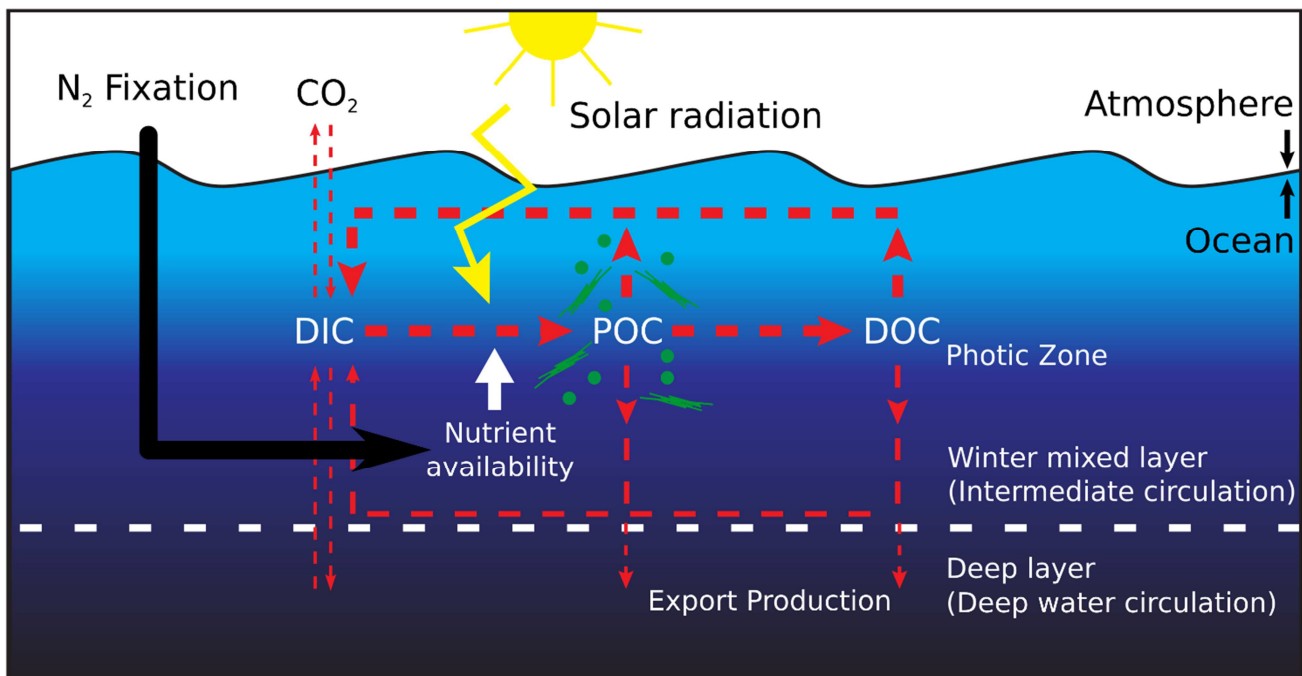

**Figure 1**

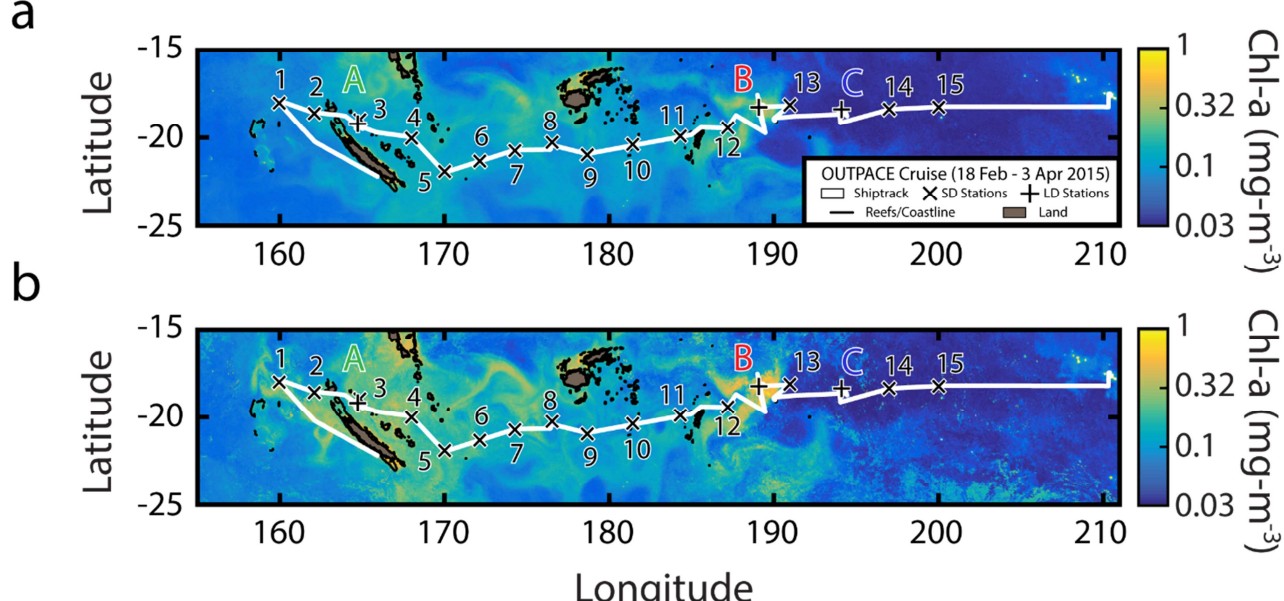

**Figure 2. a.b.**

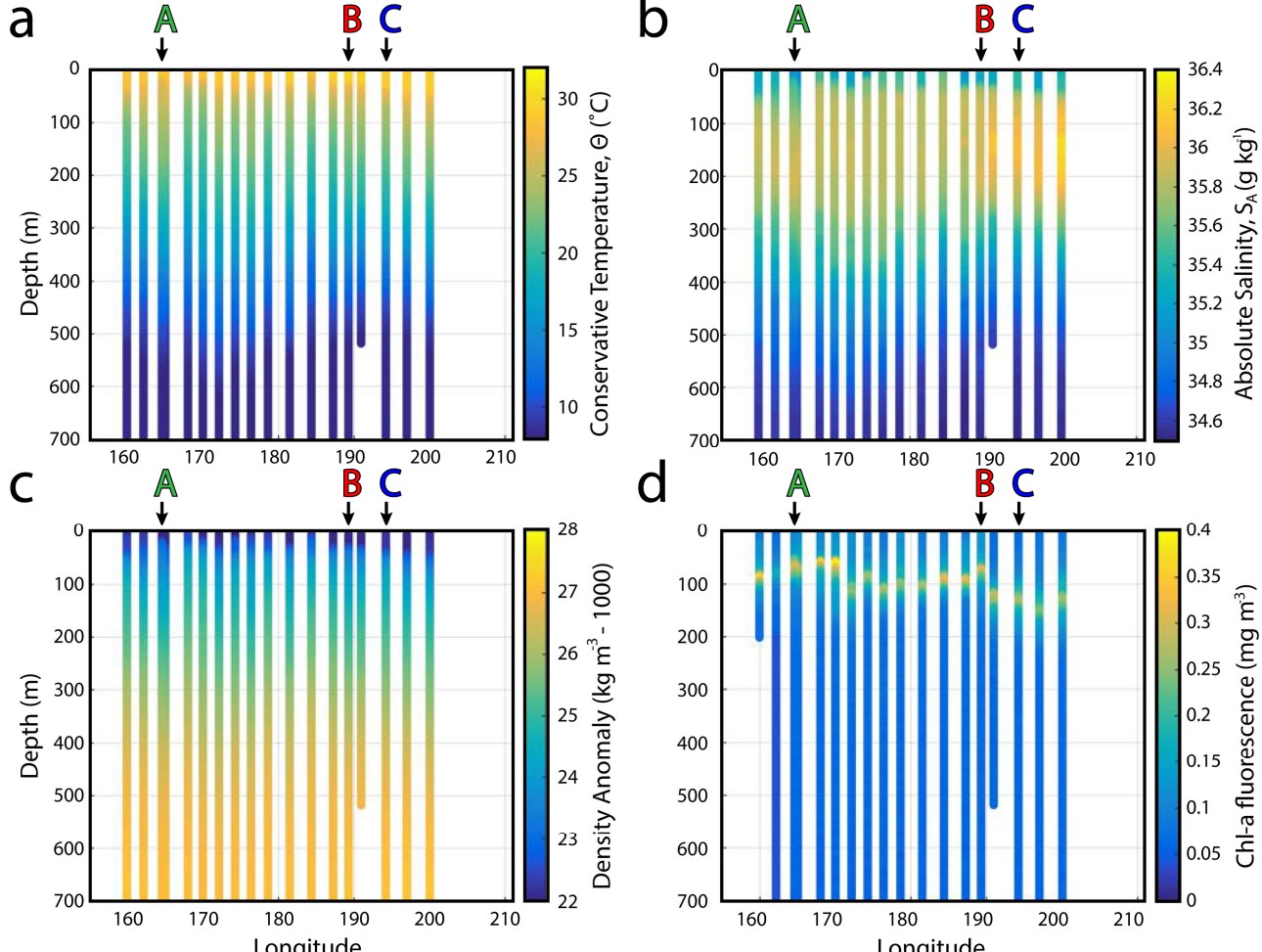

**Figure 3.a.b.c.d.**

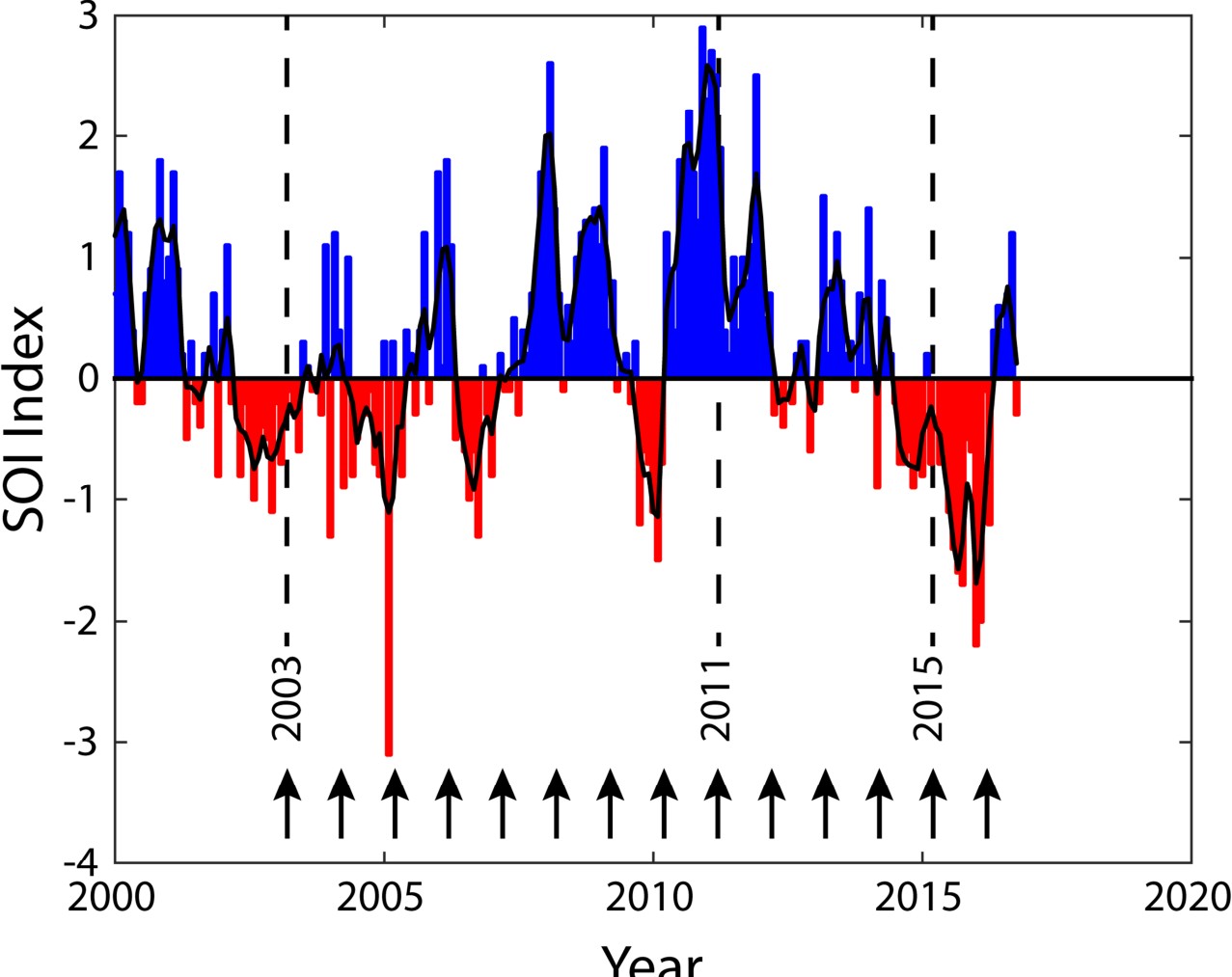

**Figure 4**

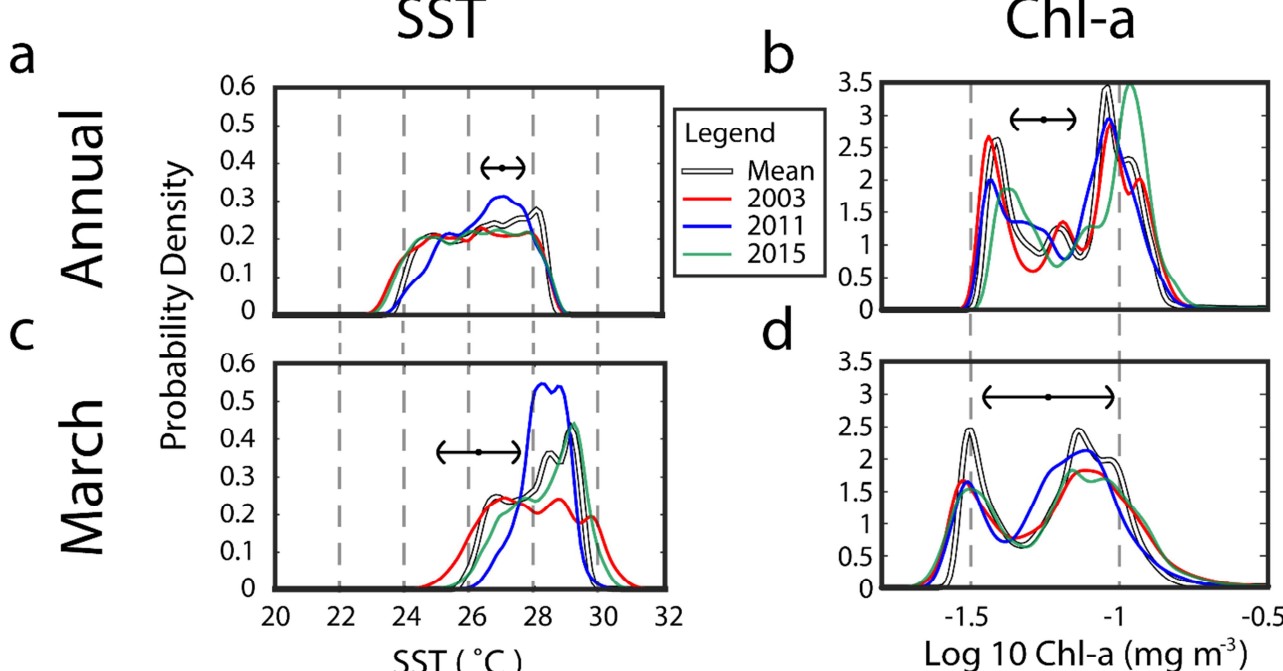

**Figure 5**