# Peer review of "The Oligotrophy to the UlTra-oligotrophy PACific Experiment (OUTPACE cruise, Feb. 18 to Apr. 3, 2015)"

_Biogeosciences, 2017_

## Referee Comment (RC1) · Anonymous Referee #1 · 29 Mar 2017

Moutin et al. present this overview paper on OUTPACE cruise. It is a well written manuscript and the objectives of the cruise are discussed clearly. But I feel there is a need to list the major findings of this cruise. Results can be discussed in individual manuscripts of the special issue separately, but important findings reported in each manuscript can just be listed (with references) here. I have some minor suggestions as follows:

Page 2: line 32, oligotrophic itself means low concentration so the 'oligotrophic' word is redundant here.

Page 3: line 9, This 60% is only the surface area (not the volume), so 'ocean' could be replaced by 'surface ocean' to make explicitly clear.

[Figure]

Page 4: lines 3-5: "A d15N budget. . . . . . . . . . . . ...export production". Please check the recent work done in the Arabian Sea by (Gandhi et al. 2011; Kumar et al. 2017), who showed that the contribution of N2 fixation to export production can be up to 92%.

Page 4: line 9, 'significant contribution', provide the quantitative estimate. I am not sure though if nanoSIMS could provide that. Since the authors mentioned 'significant', it becomes important to know the quantitative amount.

Page 4: line 15, 'this question', which question?

Page 4: line 16-21, "While average. . . . . . . . . . . . .. New Caledonia". Again see (Gandhi et al. 2011; Kumar et al. 2017), who observed the highest ever rates anywhere in the world ocean. Particularly check the table (2) in (Kumar et al. 2017) that has listed all and compared all the rates – updated after (Benavides and Voss 2015).

Page 5: line 18, satisfactory does not sound proper here. It is subjective – it could be satisfactory to one person but not to others.

Page 6:, lines 5-6, revise the sentence for grammar

Page 7: line 16, Marine Video Profiler does not acronym to UVP. Do the authors mean Underwater Vision Profiler?

Table 1: (deg, min) data are just converted to degrees in the next columns, which is redundant.

Fig. 1: N2 fixation is discussed several times in the manuscript. It would be helpful to represent this process in this schematic diagram.

References

Benavides, Mar, and Maren Voss 2015 Five Decades of N2 Fixation Research in the North Atlantic Ocean. Frontiers in Marine Science 2: 1–20.

Gandhi, Naveen, Arvind Singh, S Prakash, et al. 2011 First Direct Measurements

of N2 Fixation during a Trichodesmium Bloom in the Eastern Arabian Sea. Global Biogeochemical Cycles 25(4).

Kumar, PK, A Singh, R Ramesh, and T Nallathambi 2017 N2 Fixation in the Eastern Arabian Sea: Probable Role of Heterotrophic Diazotrophs. Frontiers: Marine Science 4: 80.

Please also note the supplement to this comment:
http://www.biogeosciences-discuss.net/bg-2017-50/bg-2017-50-RC1-supplement.pdf

---

## Referee Comment (RC2) · Anonymous Referee #2 · 8 May 2017

This manuscript is intended as the introduction to a spewcial volume with results from the OUTPACE cruise. It thus has very few data, but describes the background and purpose of the cruise and lists the stations with some characteristics. As an ordinary publication of original research, this would not work, but I clearly see the need for this type of introduction to a special volume so that this type of information does not need to be repeated in each paper.

Adding more results could obviously give a more complete and interesting story, but would both "steal" points from the research papers and decrease the role of this ms as an introduction.

I conclude supporting the autors' choice of this form of the publication and have no

suggestions for improvement.

---

## Author Comment (AC1) · 1 Jun 2017

We thank Reviewer 1 for the useful comments provided and address them below.

Rev. 1: Moutin et al. present this overview paper on OUTPACE cruise. It is a well written manuscript and the objectives of the cruise are discussed clearly. But I feel there is a need to list the major findings of this cruise. Results can be discussed in individual manuscripts of the special issue separately, but important findings reported in each manuscript can just be listed (with references) here.

Resp.: Section 7 has been modified according to the suggestions.

[revised manuscript text omitted]

Rev. 1: I have some minor suggestions as follows: Page 2: line 32, oligotrophic itself means low concentration so the 'oligotrophic' word is redundant here.

Resp.: We deleted "in the oligotrophic ocean"

Page 3: line 9, This 60% is only the surface area (not the volume), so 'ocean' could be replaced by 'surface ocean' to make explicitly clear.

Resp.: We did this correction.

Rev. 1: C1 Page 4: lines 3-5: "A d15N budget. . .. . .. . .. . .. . ...export production". Please check the recent work done in the Arabian Sea by (Gandhi et al. 2011; Kumar et al. 2017), who showed that the contribution of N2 fixation to export production can be up to 92%.

Resp.: We modified the sentence :"A $\delta$15N budget performed in the mesocosms confirmed the high contribution of N2 fixation (56 %, Knapp et al., 2016) to export compared to other tropical and subtropical regions where active N2 fixation contributes 10 to 25 % to export production (e.g. Altabet, 1988; Knapp et al., 2005)" as follows : "A $\delta$15N budget performed in the mesocosms confirmed the high contribution of N2 fixation (56 %, Knapp et al., 2016) to export compared to other tropical and subtropical

regions where active N2 fixation contributes 10 to 25 % to export production (e.g. Alta-bet, 1988; Knapp et al., 2005) and exceptionally up to 92% in the Arabian Sea (Gandhi et al. 2011; Kumar et al. 2017)."

Rev. 1: Page 4: line 9, 'significant contribution', provide the quantitative estimate. I am not sure though if nanoSIMS could provide that. Since the authors mentioned 'significant', it becomes important to know the quantitative amount.

Resp.: The sentence :" the use of nanoSIMS (nanoscale Secondary Ion Mass Spec-trometry) enabled tracking the fate of 15N from both Trichodesmium (Bonnet et al., 2016b) and UCYN blooms (Berthelot et al., 2015; Bonnet et al., 2016c), and demon-strated that a significant fraction of N originating from N2 fixation is quickly transferred to non-diazotrophic plankton, in particular diatoms (i.e. efficient C exporters to depth, (Nelson et al., 1995) during Trichodesmium blooms (Bonnet et al., 2016b)." has been modified as follows : "the use of nanoSIMS (nanoscale Secondary Ion Mass Spectrom-etry) enabled tracking the fate of 15N from both Trichodesmium (Bonnet et al., 2016b) and UCYN blooms (Berthelot et al., 2015; Bonnet et al., 2016c), and demonstrated that ∼8 % of N originating from N2 fixation is quickly transferred to non-diazotrophic plankton, in particular diatoms (i.e. efficient C exporters to depth, (Nelson et al., 1995) during Trichodesmium blooms (Bonnet et al., 2016b)."

Rev. 1: Page 4: line 15, 'this question', which question?

Resp.: We replaced the sentence by "The western tropical south Pacific (WTSP) is an ideal location to study the fate of N fixed by N2 fixation"

Rev. 1: Page 4: line 16-21, "While average. . .. . .. . .. . .. . .. New Caledonia". Again see (Gandhi et al. 2011; Kumar et al. 2017), who observed the highest ever rates anywhere in the world ocean. Particularly check the table (2) in (Kumar et al. 2017) that has listed all and compared all the rates – updated after (Benavides and Voss 2015).

[Figure]

Resp.: It is written "which is in the upper range of rates reported in the global N2 fixation MAREDAT database and even surpassed its upper rates (100-1000 $\mu$mol N m-2 d-1) (Luo et al., 2012)" which is indeed the case. We nevertheless added the sentence. Very high rates have also been recently reported in the Arabian Sea (Gandhi et al. 2011; Kumar et al. 2017).

Rev. 1: Page 5: line 18, satisfactory does not sound proper here. It is subjective – it could be satisfactory to one person but not to others.

Resp.: We deleted "satisfactory"

Rev. 1: Page 6:, lines 5-6, revise the sentence for grammar

Resp.: We deleted "as follows" in the sentence: Following the planned adaptive strategy, the initial transect designed to approximately follow 19° S was modified along-route thanks to the information coming from satellite images.

Rev. 1: Page 7: line 16, Marine Video Profiler does not acronym to UVP. Do the authors mean Underwater Vision Profiler?

Resp.: Yes, thank you.

Rev. 1: Table 1: (deg, min) data are just converted to degrees in the next columns, which is redundant.

Resp.: Right but it is helpful for the other scientists who will publish in the special issue. We prefer to leave it like it is to avoid future conversion error in other ms.

Rev. 1: Fig. 1: N2 fixation is discussed several times in the manuscript. It would be helpful to represent this process in this schematic diagram.

Resp.: We added N2 fixation in Figure 1.

[revised manuscript text omitted]

Please also note the supplement to this comment:
http://www.biogeosciences-discuss.net/bg-2017-50/bg-2017-50-AC1-supplement.pdf

―――――――――――――――

[Figure]

N₂ Fixation    CO₂       Solar radiation              Atmosphere

DIC        POC        DOC    Photic Zone

Nutrient
availability

Winter mixed layer
(Intermediate circulation)

Export Production    Deep layer
(Deep water circulation)

Ocean

**Fig. 1.**

---

## Author Comment (AC2) · 1 Jun 2017

Reviewer 2: This manuscript is intended as the introduction to a special volume with results from the OUTPACE cruise. It thus has very few data, but describes the background and purpose of the cruise and lists the stations with some characteristics. As an ordinary publication of original research, this would not work, but I clearly see the need for this type of introduction to a special volume so that this type of information does not need to be repeated in each paper.

Adding more results could obviously give a more complete and interesting story, but would both "steal" points from the research papers and decrease the role of this ms as an introduction.

I conclude supporting the autors' choice of this form of the publication and have no suggestions for improvement.

Response: We thank Reviewer 2 and appreciate his support of this form of publication as an introduction to our special volume.